# Inflammatory and Oxidative Biological Profiles in Mental Disorders: Perspectives on Diagnostics and Personalized Therapy

**DOI:** 10.3390/ijms26199654

**Published:** 2025-10-03

**Authors:** Izabela Woźny-Rasała, Ewa Alicja Ogłodek

**Affiliations:** Collegium Medicum, Jan Dlugosz University in Częstochowa, Waszyngtona 4/8 Street, 42-200 Częstochowa, Poland; e.oglodek@ujd.edu.pl

**Keywords:** laboratory diagnostics, neuroinflammatory biomarkers, oxidative stress, neuroplasticity, personalized psychiatry, precision psychiatry, pro-inflammatory cytokines

## Abstract

Personalized psychiatry represents an innovative therapeutic approach that integrates biological, genetic, and clinical data to optimize the treatment of mental disorders. Laboratory diagnostics play a fundamental role in this process by providing precise biomarkers that characterize pathophysiological mechanisms such as neuroinflammatory processes, oxidative stress, dysfunction of the Hypothalamic–Pituitary–Adrenal (HPA) axis, as well as disturbances in neuroplasticity and neurodegeneration. This article discusses the use of advanced analytical techniques, such as immunoenzymatic assays for pro-inflammatory cytokines (Interleukin-1β- IL-1β; Interleukin-6-IL-6; Interleukin-18-IL-18; and Tumor Necrosis Factor- α - TNF-α). It also emphasizes the role of pharmacogenomic diagnostics in the individualization of psychotropic therapy. Interdisciplinary collaboration between laboratory diagnosticians and clinicians supports the potential for multidimensional analysis of biomarker data in a clinical context, which supports precise patient profiling and monitoring of treatment responses. Despite progress, there are limitations, such as the lack of standardization in measurement methods, insufficient biomarker validation, and limited availability of tests in clinical practice. Development prospects include the integration of multi-marker panels, the use of point-of-care diagnostics, and the implementation of artificial intelligence tools for the analysis of multidimensional data. As a result, laboratory diagnostics are becoming an integral element of personalized psychiatry, enabling a better understanding of the neurobiology of mental disorders and the implementation of more effective therapeutic strategies.

## 1. Introduction

In recent years, a groundbreaking shift has been observed in the approach to diagnosing and treating mental disorders—from the traditional model based on describing clinical symptoms toward the concept of personalized psychiatry, where the identification of molecular and biological pathophysiological mechanisms plays a central role [1,2]. The diagnostic classification included in the *Diagnostic and Statistical Manual of Mental Disorders, Fifth Edition (DSM-5)* remains a fundamental clinical tool, but its limitations in capturing the neurobiological phenotypic diversity of mental disorders are increasingly highlighted in the scientific literature [3]. Disorders such as Major Depressive disorder (MDD), anxiety, Post-Traumatic Stress Disorder (PTSD), or schizophrenia are characterized by multifactorial etiology and complex pathogenesis, involving interactions among neuroinflammatory, neuroendocrine, epigenetic, neurotransmitter, and metabolic mechanisms that significantly exceed the scope of classical diagnostic criteria [4,5,6,7]. In response to these challenges, there is growing interest in identifying the so-called biological endophenotypes—defined not only by observable clinical symptoms but also by molecular, physiological, and bioinformatic profiles. These endophenotypes form a crucial link between genotype and clinical phenotype, reflecting, among other things, patterns of cytokine and chemokine expression, dysregulation of the Hypothalamic–Pituitary–Adrenal (HPA) axis, Deoxyribonucleic Acid (DNA) methylation changes, and biomarkers of oxidative stress, inflammasome activation, neurotransmitter disturbances, or neurodegenerative processes [8,9,10]. Their identification may significantly improve diagnostic accuracy and sensitivity, allowing for more precise differentiation of symptom clusters, prediction of treatment response, and reduction in relapse risks. In this context, diagnostic laboratories are becoming not only the technological backbone of clinical medicine but also active participants in the diagnostic process, providing high-resolution molecular data essential for realizing the concept of precision psychiatry [11,12]. Modern laboratories are equipped with advanced analytical technologies, such as multiparametric flow cytometry, Liquid Chromatography–tandem Mass Spectrometry (LC-MS/MS), quantitative Polymerase Chain Reaction (qPCR), Western blotting, and multiplex immunoenzymatic assays (e.g., ELISA). These tools allow for precise measurement of a range of neuroinflammatory, oxidative, and neurodegenerative biomarkers, as well as signaling molecules relevant to understanding the biology of mental disorders [13,14,15].

Special attention is given to the role of cytokines and chemokines which act as central mediators of neuroinflammatory processes in MDD, PTSD, and schizophrenia [16,17,18,19].

Persistent inflammation in the central nervous system, linked to microglial activation, astrocytic dysfunction, and blood–brain barrier impairment, contributes to altered neurotransmission and the maintenance of depressive, anxiety, and cognitive symptoms [20,21]. Nevertheless, study findings remain heterogeneous. Variations in cytokine levels may arise from patient-related factors—such as age, sex, comorbidities, and medication use—as well as methodological differences, including assay sensitivity and sample type.

Cytokine profiling enables not only the differentiation of inflammatory and non-inflammatory phenotypes of mental disorders but also the assessment of individual susceptibility to disorders associated with oxidative stress and mitochondrial dysfunction. An important supplement to these analyses includes measurements of biomarkers such as Malondialdehyde (MDA), Ubiquitin Carboxy-Terminal Hydrolase L1 (UCHL1), Small Ubiquitin-Related Modifier 1 (SUMO1), Caspase-1, and Activating Transcription Factor 6 (ATF6)—indicators of key factors in the oxidative–inflammatory stress axis, endoplasmic reticulum damage, and activation of the Unfolded Protein Response (UPR), whose dysregulation is significant in the pathophysiology of affective and neurodegenerative disorders [22,23,24]. Omics techniques are also developing in parallel, enabling global analysis from the genome level through the transcriptome, proteome, and metabolome to the epigenome. Genomics provides information about genetic variants related to susceptibility to mental disorders, for example, within genes associated with neurotransmitter receptor function, regulators of the HPA axis, cytokine pathway proteins, or glutamate transport. Transcriptomics allows the analysis of dynamic gene expression changes in response to psychological stress, while proteomics and metabolomics provide data on the functional effects of these changes, including enzyme levels, receptors, inflammatory mediators, and neuroactive metabolites in body fluids such as serum, plasma, or cerebrospinal fluid [25,26,27,28]. The integrative “multi-omics” approach is gaining importance in translational psychiatry, enabling the correlation of data from various levels of biological organization—from genetic information to the metabolic phenotype—to map complex pathobiological networks [29,30]. Epigenomics, including research on DNA methylation and histone modifications, offers new perspectives on understanding the lasting effects of stress. Although the genome sequence itself is not altered, epigenetic mechanisms can lead to long-term changes in the expression of genes responsible for mood regulation, neuroplasticity, microglial activity, and inflammatory responses. Analysis of these phenomena allows the identification of individuals particularly vulnerable to developing mental disorders and serves as a tool to assess the effectiveness of pharmacological and psychotherapeutic treatments. The application of modern molecular technologies in psychiatric research facilitates the detection of subclinical pathological processes that remain invisible within classical diagnostic frameworks [31,32,33]. This approach paves the way for the construction of integrated pathobiological models, allowing for precise mapping of signaling pathways and molecular networks disrupted in specific disease entities. Combining these data with neuroimaging results, microbiome analysis, epigenetic profiles, and clinical data provides a solid foundation for the development of precision psychiatry, which is focused on identifying specific molecular mechanisms rather than merely treating symptoms. In light of these trends, the role of laboratory diagnosticians is undergoing a profound transformation [34,35,36]. They are becoming not only a performer of laboratory assays but also an active member of an interdisciplinary clinical-research team, integrating molecular data with the patient’s clinical profile [37]. The following sections of this paper will present the current capabilities of laboratory diagnostics in personalized psychiatry, including the assessment of the pro- and anti-inflammatory cytokine balance and oxidative stress markers, the calculation of relational inflammatory indices (e.g., the Interleukin-6/Interleukin-10 ratio - IL-6/IL-10 ratio), as well as the clinical significance of identifying inflammatory phenotypes among patients with mental disorders.

The aim of this work is to demonstrate the potential of molecular diagnostics as an integral element of modern psychiatry—based on biologically justified patient stratification—allowing for individualized treatment selection and early identification of individuals at risk of a chronic disease course. Personalized psychiatry, in which laboratory diagnostics plays an increasingly important role, is a response to contemporary clinical challenges, setting new directions for the development of neuropsychiatry and precision medicine.

## 2. Balance of Pro- and Anti-Inflammatory Cytokines as a Pathobiological Axis in Affective, Anxiety, and Schizophrenic Disorders

Traditional diagnostic methods, which are based mainly on clinical interviews and symptom observation, are increasingly being supplemented with laboratory tests that identify biological biomarkers. Among these, special attention is given to the assessment of the immune system, particularly the balance between pro- and anti-inflammatory cytokines, which play a significant role in the pathophysiology of many psychiatric disorders [38,39]. Personalized psychiatry assumes that each patient with a mental disorder has a unique biological profile that determines the course of the illness and the response to treatment. In clinical practice, this means the need to integrate various types of data, including molecular and biochemical, which may help identify the disease profile, predict its progression, and guide pharmacotherapy decisions [40,41]. Laboratory diagnostics support the application of modern methods to analyze biological markers in blood serum, cerebrospinal fluid, and genetic material. Contemporary neuropsychiatry increasingly draws on knowledge from molecular immunology to better characterize the basis of anxiety, MDD, bipolar disorder, and schizophrenia [42,43]. At the center of these studies are cytokines and chemokines, which serve as modulators of synaptic plasticity and neurogenesis (Table 1).

### 2.1. Pro-Inflammatory Cytokines in the Pathophysiology of Psychiatric Disorders

Interleukin 6 (IL-6) is a classic example of a pro-inflammatory factor with key im-portance in regulating the Janus Kinase/Signal Transducer and Activator of Transcription 3 (JAK/STAT3) pathway, inducing acute-phase proteins, and shaping monoaminergic activity in the central nervous system. Numerous studies show that high serum levels of IL-6 predict persistent depressive symptoms and poor responses to Selective Serotonin Reuptake Inhibitors (SSRIs), whereas a decrease in IL-6 correlates with symptom remission, making this cytokine both a prognostic and pharmacodynamic biomarker [44,45,93].

Moreover, IL-6 also plays a significant role in the pathogenesis of neuropathic pain, which often accompanies depressive episodes. Through the activation of glial cells and modulation of the JAK/STAT3 pathway, this cytokine contributes to central sensitization, leading to the persistence of chronic pain stimuli. Experimental models have shown that inhibition of the IL-6 receptor reduces neuropathic symptoms, while in patients with depression, higher IL-6 levels are associated with increased pain severity and poorer responses to antidepressant treatment [94]. This suggests that IL-6 may link inflammatory mechanisms underlying both depressive and pain symptoms. Therefore, it represents an important biomarker and a potential therapeutic target for interventions addressing psychiatric as well as neurological disorders [95].

Interleukin 1β (IL-1β) activates the NOD-like Receptor Family and the Pyrin Domain containing 3 (NLRP3) inflammasome, increases blood–brain barrier permeability, and reduces Brain-Derived Neurotrophic Factor (BDNF) expression in the hippocampus. In bipolar disorder, elevated IL-1β levels have been observed during the manic phase, indicating the neurotoxic effects of glutamate [96]. In schizophrenia, early elevation of IL-1β correlates with negative symptoms, showing its relevance in the preventive diagnosis of negative symptoms and associated cognitive deficits, yet further studies are needed to confirm this association [97].

Interleukin-18 (IL-18), a member of the interleukin-1 (IL-1) family, activates the Myeloid Differentiation primary response 88/Nuclear Factor kappa-light-chain-enhancer of activated B cells (MyD88/NF-κB) pathway, enhancing Interferon- γ (IFN-γ) production and amplifying the inflammatory response. In PTSD, elevated IL-18 levels correlate with hippocampal atrophy and intensified intrusive symptoms, indicating its role in neuroinflammatory pathogenesis. In MDD, IL-18 contributes to chronic inflammation, inhibiting neurogenesis and neuroplasticity, which are associated with symptom severity and treatment resistance. In bipolar disorder, increased IL-18 is linked to intensified neuroinflammatory processes and cognitive dysfunction. In schizophrenia, elevated IL-18 correlates with progressive neurodegeneration and the worsening of negative and cognitive symptoms, confirming its involvement in neuroinflammatory dysregulation. IL-18 is thus an important pathophysiological factor and biomarker across a broad spectrum of neuropsychiatric disorders [73].

IFN-γ is a cytokine mainly produced by T helper type 1 (Th1) lymphocytes, Natural Killer cells (NK cells), and cytotoxic Cluster of Differentiation 8-Positive T Lymphocytes (CD8+) T cells, playing a central role in cell-mediated immune responses. IFN-γ induces the expression of Major Histocompatibility Complex (MHC) class I and II molecules, activates macrophages, and stimulates the production of chemokines such as CXCL9, CXCL10, and CXCL11, which drive effector cell migration to sites of inflammation. A dominance of IFN-γ signaling over Interleukin-4 (IL-4) (a high IFN-γ/IL-4 ratio) indicates a Th1-dominant immune response and serves as a marker of pro-inflammatory polarization. In schizophrenia, there is a shift in immune balance toward Th1 pathway dominance, as reflected by elevated IFN-γ levels, which correlate with increased negative symptoms, social anergy, and cognitive deficits [98].

These findings support the notion that Th1 axis hyperactivity contributes to the pathophysiology of treatment-resistant schizophrenia subtypes and highlight the potential of α7 nicotinic receptor (α7 nAChR) agonists as a therapeutic strategy, given their ability to modulate IFN-γ expression and restore the neuroimmune balance through anti-inflammatory effects [99]. In MDD, increased IFN-γ activity is associated with resistance to serotonergic treatment and exacerbation of somatic symptoms, anhedonia, and dysphoria. IFN-γ may also influence the Indoleamine 2,3-dioxygenase (IDO)–kynurenine pathway, promoting neurotoxic metabolites (e.g., quinolinic acid) and further disrupting glutamatergic neurotransmission. In bipolar disorder, elevated IFN-γ levels are observed during depressive and mixed episodes, correlating with circadian rhythm disturbances, psychotic symptoms, and a high suicide risk [46]. In patients with bipolar disorder, IFN-γ may also affect responses to mood stabilizers, particularly lithium, which partially suppresses the Th1 response [46]. In anxiety disorders (including generalized anxiety and PTSD), high IFN-γ levels indicate immune activation and correlate with hypercortisolemia, sleep disturbances, and intensified somatic symptoms. IFN-γ may modulate neuronal plasticity and neurogenesis in the hippocampus, contributing to the persistence of anxiety symptoms and emotional disturbances [100]. Interferon gamma constitutes a significant common element in the immunopathology of MDD, bipolar disorder, schizophrenia, and anxiety disorders. Its role as a biomarker of inflammatory responses and a potential therapeutic target (e.g., via α7nAChR, modulation of the kynurenine pathway or anti-cytokine antibodies) warrants further research. IFN-γ is a pro-inflammatory cytokine produced primarily by Th1 cells, NK cells, and cytotoxic CD8+ T lymphocytes, playing a central role in the cell-mediated immune response. IFN-γ induces the expression of MHC class I and II molecules, enhances antigen presentation, activates macrophages, and promotes the production of CXCL family chemokines (particularly CXCL9, CXCL10, and CXCL11), which mediate T lymphocyte recruitment to inflammation sites. A high IFN-γ-to-IL-4 ratio indicates a Th1-type response associated with chronic immune activation [47]. In schizophrenia, excessive Th1 axis activity and elevated IFN-γ levels correlate with the severity of negative symptoms such as social withdrawal [101]. This may result from chronic neuroinflammatory microglial activation induced by Th1 cytokines, leading to damage in frontolimbic pathways and disrupted synaptic plasticity. In this context, α7 nAChR agonists—by modulating pro-inflammatory cytokine release and exerting neuroprotective effects—represent a promising therapeutic strategy, pending further validation. In MDD, especially with an inflammatory profile, IFN-γ plays a significant role by activating the IDO pathway, leading to the conversion of tryptophan into neurotoxic kynurenine metabolites, such as quinolinic acid. This mechanism reduces serotonin availability and disrupts N-methyl-D-aspartate (NMDA)–glutamatergic signaling, contributing to anhedonia, sleep disturbances, and somatic symptoms. Increased IFN-γ expression contributes to resistance to standard antidepressants and underscores the potential need to investigate immunomodulatory therapies [48].

In bipolar disorder, IFN-γ levels are particularly elevated during depressive and mixed phases, and this increase may correlate with affective symptom severity, circadian rhythm disturbances, and relapse susceptibility [102]. IFN-γ may also influence neurogenesis and cognitive function stability, particularly via interactions with the kynurenine and neuroendocrine systems. In anxiety disorders, including Generalized Anxiety Disorder (GAD) and PTSD, IFN-γ is elevated in some patients and is associated with GABAergic system dysfunction. Its activity can disrupt neuroimmune homeostasis, leading to persistent anxiety symptoms and sleep disturbances. In PTSD, IFN-γ levels also correlate with the intensity of flashbacks and disturbances in emotional memory integration [51]. In summary, IFN-γ plays a role in the pathophysiology of many psychiatric disorders with an inflammatory component—including MDD, Bipolar Disorder (BD), schizophrenia, and anxiety disorders—both as an effector factor of the Th1 response and as a mediator of neurotoxic metabolic pathways. Its level may serve as an inflammatory biomarker, a predictor of therapeutic responses, and a potential target for pharmacological intervention in immunomodulatory treatment strategies.

Tumor Necrosis Factor alpha (TNF-α) is a pro-inflammatory cytokine that plays a central role in the pathogenesis of various neuropsychiatric disorders, including MDD, schizophrenia, and bipolar disorder [103]. Through interactions with Tumor Necrosis Factor Receptor 1 (TNFR1) and Tumor Necrosis Factor Receptor 2 (TNFR2), TNF-α initiates signaling cascades with diverse effects—from neuronal apoptosis (TNFR1 pathway) to neuroprotective and regenerative processes (TNFR2 pathway). In affective disorders, particularly treatment-resistant MDD, TNFR1 pathway dominance leads to caspase activation (including caspase-8 and -3), resulting in synaptic degeneration, impaired neuroplasticity, and atrophy of the hippocampus and prefrontal cortex [104]. TNF-α also affects glutamatergic neurotransmission through modulation of α-amino-3-hydroxy-5-methyl-4-isoxazolepropionic acid (AMPA) receptors. Increased Glutamate Receptor 1 subunit AMPA-type (GluA1 AMPA receptor) expression induced by TNF-α enhances postsynaptic excitation and contributes to excitotoxicity. This imbalance within the cortico-limbic axis may lead to dysfunction in networks regulating mood, motivation, and cognitive functions, which are especially relevant in the pathophysiology of treatment-resistant MDD, where chronic TNF-α activation correlates with a poor antidepressant response and reduced neurogenesis [105]. In schizophrenia, TNF-α also plays an important role. Elevated TNF-α levels in plasma and cerebrospinal fluid are observed particularly in patients with prominent negative symptoms and cognitive impairments [81]. TNF-α modulates synaptic plasticity and glutamatergic neurotransmission, as well as affects astrocyte and microglial function, disrupting the neurochemical balance in mesolimbic and frontal-cortical pathways [52]. Additionally, TNF-α can influence intracellular gene expression changes involved in NMDA receptor function [106]. In BD, the effects of TNF-α vary depending on the illness phase. In the depressive phase, increased serum levels and activation of pro-inflammatory signaling pathways are observed, while in the manic phase, neuroprotective mechanisms associated with TNFR2 may dominate [53]. TNF-α impacts intracellular signaling pathways such as nuclear factor kappa-light-chain-enhancer of activated B cells (NF-κB) and Mitogen-Activated Protein Kinase (MAPK), and its expression correlates with circadian rhythm disturbances and the severity of depressive symptoms. It contributes to disrupted synaptic homeostasis, changes in glutamatergic signaling, and neuroplasticity deficits [54]. Moreover, TNF-α is a potential therapeutic target, either through direct inhibition (e.g., TNF-α blockers) or indirect modulation of inflammatory responses (e.g., targeting microglia or caspase and NF-κB signaling pathways) [55].

Interleukin-8, also widely known as C-X-C motif chemokine ligand-8 (IL-8, CXCL8), stimulates microglial migration and activation and influences the dynamics of neuroinflammatory neuron–glia interactions. In first-episode psychosis, elevated peripheral IL-8 levels correlate with an unfavorable clinical course and an increased risk of progression to treatment-resistant schizophrenia, indicating that IL-8 may serve as a predictive biomarker and supporting its early monitoring in the differential diagnosis of psychotic disorders [58]. In bipolar disorder, altered IL-8 expression is observed in both the depressive and manic phases, with heightened inflammatory axis activation associated with intensified affective symptoms and impaired regulation of mood and cognition [59]. IL-8 may affect synaptic plasticity and the blood–brain barrier, contributing to the consequences of chronic microglial activation and potentially contributing to neuro-degenerative mechanisms in BD [60]. In major depressive disorder, especially in treatment-resistant subtypes, significantly elevated IL-8 levels are found in serum and cerebrospinal fluid. Increased IL-8 levels are associated with more severe depressive symptoms, sleep disturbances, anhedonia, and resistance to pharmacological treatments. The neurotoxic cytokine profile involving IL-8 may lead to persistent impairment of hippocampal neurogenesis and dysregulation of the HPA axis, further worsening emotional disturbances [61]. From a transdiagnostic perspective, IL-8 emerges as a shared pathophysiological factor across psychiatric disorders with an inflammatory component, with its expression profile potentially distinguishing the disease course and prognosis in schizophrenia, bipolar disorder, and major depressive disorder [62]. Early identification of elevated IL-8 levels may therefore support clinical prognoses and provide a rationale for considering targeted immunomodulatory interventions [82].

Pro-inflammatory cytokines such as IL-6, IL-1β, TNF-α, and IL-8 contribute to neuroinflammation, synaptic dysfunction, and impaired neuroplasticity, thereby playing a key role in the onset and progression of psychiatric disorders.

### 2.2. Anti-Inflammatory and Immunoregulatory Cytokines in Neuropsychiatry

Interleukin 10 has anti-inflammatory properties and can suppress the expression of pro-inflammatory cytokines (including IL-1β, IL-6, and TNF-α), chemokines, and cell adhesion molecules. It acts mainly through activation of the Janus Kinase 1/Signal Transducer and Activator of Transcription 3 (JAK1/STAT3) signaling pathway and indirectly by promoting the differentiation of microglia toward the M2 phenotype—a neuroprotective type associated with tissue repair, neurogenesis, and restoration of synaptic homeostasis. The action of IL-10 is based primarily on inducing the JAK1/STAT3 signaling pathway, leading to modulation of gene expression involved in the anti-inflammatory response. This results in activation of M2 phenotype microglia, which perform neuroprotective functions, facilitate neural tissue repair, promote neurogenesis, and help restore and maintain synaptic homeostasis. In MDD, particularly with sleep disturbances, studies have shown significantly reduced IL-10 levels in both plasma and cerebrospinal fluid, which show intensified neurotoxic activity of M1-type microglia. Low IL-10 activity correlates with reduced expression of neurotrophic factors (e.g., BDNF), increased susceptibility to anhedonia, circadian rhythm disruptions, and resistance to pharmacological treatments. For this reason, interventions aimed at increasing IL-10 levels—such as the use of omega-3 fatty acids or anti-inflammatory diets—are gaining interest in translational psychiatry. In anxiety disorders, including generalized anxiety disorder and PTSD, reduced IL-10 expression is associated with persistent low-grade inflammation, which may lead to dysregulation of the limbic system (especially the amygdala and anterior cingulate cortex) and emotional disturbances [63]. IL-10 may also indirectly influence the activity of the HPA axis by modulating glucocorticoid receptor sensitivity and reducing stress susceptibility. In bipolar disorder, the IL-10 profile depends on the illness phase [107]. During the manic phase, relatively higher IL-10 levels are observed compared to the depressive phase, which may indicate a temporary regulatory mechanism suppressing excessive inflammatory activation. However, long-term IL-10 deficiencies in BD patients are associated with chronic illness progression, more severe cognitive deficits, and a poorer response to mood stabilizers, indicating involvement of this cytokine in neurodegenerative mechanisms and glutamatergic neurotransmission disturbances [108]. In schizophrenia, data indicate reduced IL-10 levels in serum and cerebrospinal fluid, especially in cases with dominant negative symptoms and treatment resistance [109]. Weakened anti-inflammatory responses promote sustained activation of M1-type pro-inflammatory microglia, resulting in disrupted development of connections between the cortex and subcortical structures and impaired synaptic pruning during brain maturation [110]. IL-10 also plays an important role by influencing dopaminergic and GABAergic neurotransmission. It modulates the activity of these neurochemical systems, which are essential for regulating cognitive functions and controlling psychotic symptoms. As a result, changes in IL-10 levels may affect the severity of psychotic symptoms, such as delusions or hallucinations, as well as cognitive deficits often observed in psychiatric disorders, including schizophrenia and affective disorders. Due to this complex role, IL-10 is gaining significance as a potential biomarker for monitoring both clinical status and the effectiveness of immunomodulatory therapies [64]. Such therapies include the use of minocycline, low-dose aspirin, or dietary interventions based on anti-inflammatory substances (e.g., omega-3 fatty acids) and aim to modulate the immune system and reduce the neuroinflammatory component of psychiatric disorder pathogenesis. Monitoring IL-10 levels may therefore provide valuable insights into the effectiveness of these therapies and support a more personalized therapeutic approach [111].

Transforming growth factor β (TGF-β) plays a key role in regulating immunomodulation and neurobiological homeostasis, primarily through the induction of Forkhead Box P3 (FOXP3) expression and promotion of the regulatory T lymphocyte (Treg) phenotype. In anxiety disorders, reduced TGF-β levels correlate with amygdala hyperactivity and deficits in cortico-limbic regulation, while normalization of its levels after exposure therapy is associated with improved emotional functioning. In MDD, decreased TGF-β activity contributes to a chronic neuroinflammatory state, limiting hippocampal neurogenesis and neuroplasticity, which correlate with symptom severity and treatment resistance [56]. In bipolar disorder, TGF-β dysregulation affects mood variability by modulating neuroinflammatory and neuroplastic pathways in the prefrontal cortex and hippocampus [66]. In schizophrenia, TGF-β deficiency is linked to increased cognitive dysfunction and negative symptoms, serving as a marker of a chronic neuroinflammatory phenotype and reduced neuroprotection [68]. Thus, modulation of TGF-β signaling represents a promising direction in the treatment of neuropsychiatric disorders, aiming to restore the immuno-neurobiological balance and improve brain function.

C-X3-C motif chemokine ligand 1 (CX3CL1, fractalkine) is a chemokine that plays an important role in neuron–microglia communication via its specific binding to the C-X3-C Motif Chemokine Receptor 1 (CX3CR1) receptor present on the surface of microglia. In the pathophysiology of neuropsychiatric disorders, reduced CX3CL1 expression reflects impaired neuron–microglia interactions, resulting in intensified inflammation and disrupted neuroplasticity [69]. In schizophrenia, decreased CX3CL1 levels correlate with heightened expression of negative symptoms such as apathy, anhedonia, and cognitive deficits, reflecting weakened neuroprotective microglial function [70]. Interestingly, increased CX3CL1 levels following clozapine treatment show that fractalkine may serve as a biomarker of response to atypical antipsychotics, associated with restored neuroinflammatory balance and synaptic function. In MDD, CX3CL1 plays a role in modulating neuroinflammatory processes and neurogenesis, particularly in the hippocampus and prefrontal cortex. A reduction in this chemokine contributes to chronic M1-type microglial activation and inhibition of repair processes, correlating with increased depressive symptoms such as anhedonia and reduced motivation [112]. In bipolar disorder, dysregulation of the CX3CL1–CX3CR1 axis is linked to mood variability and both manic and depressive episodes. Abnormal fractalkine signaling may disrupt neuroinflammatory balance and synaptic plasticity, affecting affective stability [71]. In anxiety disorders, CX3CL1 influences emotional reactivity by modulating microglial activity. Reduced fractalkine levels are associated with increased stress sensitivity and heightened anxiety symptoms [72].

IDO is an enzyme initiating tryptophan catabolism along the kynurenine pathway, converting tryptophan into kynurenine. Increased IDO activity reduces tryptophan availability for serotonin synthesis, leading to lower levels of this neurotransmitter in the central nervous system. Simultaneously, elevated levels of neurotoxic kynurenine pathway metabolites contribute to glutamatergic homeostasis disruption, worsening neurodegeneration and synaptic dysfunction.

The kynurenine/tryptophan (KYN/TRP) ratio, measured in serum or plasma using chromatographic methods (High Performance Liquid Chromatography-HPLC or LC-MS/MS), serves as a sensitive indicator of this process. Elevated KYN/TRP values are associated with more severe depressive symptoms, resistance to serotonergic treatment, and impaired cognitive function. In turn, meta-analyses, including Bartoli et al. [75], support its potential as a biomarker for differentiating inflammatory depression phenotypes and monitoring the effects of pharmacological therapy.

In MDD, an imbalance between neurotoxic and neuroprotective metabolites (e.g., kynurenine, kynurenine 3-monooxygenase) is closely associated with intensified inflammatory responses and decreased hippocampal neuroplasticity. Increased IDO activity and a higher kynurenine/tryptophan ratio correlate with greater symptom severity, particularly in patients with immune-driven MDD and resistance to standard serotonergic treatment [76].

The meta-analyses by Bartoli et al. [75] reveal that alterations in the kynurenine pathway may vary across psychiatric disorders. In MDD, reduced tryptophan levels are most consistently observed, which may limit serotonin synthesis and contribute to symptom severity. In bipolar disorder, lower levels of tryptophan, kynurenine, and kynurenic acid shift the pathway toward neurotoxic metabolites. In schizophrenia, there is some evidence of reduced tryptophan and kynurenic acid, although findings on the KYN/TRP ratio—which reflects IDO activity—are less consistent and may depend on the illness phase or treatment status. The authors propose that markers such as KYN/TRP and KA/KYN could serve as potential biomarkers for distinguishing inflammatory from non-inflammatory phenotypes and for monitoring responses to anti-inflammatory or glutamatergic-modulating therapies.

In bipolar disorder, activation of the kynurenine pathway and elevated IDO expression are associated with episodes of exacerbation, increased cognitive deficits, and heightened neuroinflammation, highlighting the relevance of tryptophan metabolism in the pathophysiology of the disorder and its potential therapeutic targets through immune modulation [77]. In anxiety disorders, increased IDO activity may exacerbate anxiety symptoms by disturbing the balance between neuroprotective and neurotoxic kynurenine metabolites, leading to dysregulation of serotonergic and glutamatergic systems [78]. In schizophrenia, increased IDO activity and the shift in tryptophan metabolism toward neurotoxic metabolites are linked to more severe negative symptoms, cognitive deficits, and progressive neurodegeneration.

Interleukin 2 (IL-2), a cytokine that stimulates proliferation of Cluster of Differentiation 4-Positive T Lymphocytes (CD4+) and CD-8-positive T lymphocytes (CD8+), plays an essential role in maintaining immune homeostasis and balancing effector and regulatory immune responses. In neuropsychiatric disorders, IL-2 dysregulation reflects immune system activation or insufficiency [65]. In schizophrenia, reduced IL-2 levels reflect T lymphocyte dysfunction, correlating with increased negative symptoms and cognitive deficits, and may serve as a marker of chronicity and treatment resistance [79]. In MDD, an increase in IL-2 after ketamine administration indicate a restoration of immune balance and potential involvement of this cytokine in rapid therapeutic responses, especially in patients with an inflammatory phenotype [80]. In bipolar disorder, IL-2 level fluctuations accompany different illness phases, and its reduction may be associated with impaired immune regulation and inflammatory neurotoxicity [113]. In anxiety disorders, IL-2 deficiency may contribute to sustained hyperactivity of the HPA axis and chronic stress reactivity, supporting the concept of low IL-2 as a marker of neuroimmune dysregulation [114]. IL-2 is thus a potential biomarker of clinical status and response to immune-modulating therapies, with diagnostic and prognostic significance in MDD, bipolar disorder, schizophrenia, and anxiety disorders.

Interleukin-4 is a cytokine characteristic of the Th2-type immune response, playing a role in regulating anti-inflammatory processes. Mainly produced by Th2 lymphocytes, mast cells, and basophils, IL-4 induces the expression of the transcription factor GATA-binding protein 3 (GATA3) and stimulates B lymphocyte proliferation and class switching towards Immunoglobulin E (IgE) and Immunoglobulin G subclass 4 (IgG4) antibodies. At the same time, it inhibits Th1 lymphocyte differentiation and the production of pro-inflammatory cytokines such as IFN-γ and interleukin-12 (IL-12), making it a key factor in balancing the inflammatory response and limiting toxic activity in the central nervous system. In MDD, a significant reduction in IL-4 levels is observed in both plasma and cerebrospinal fluid, showing a dominance of the Th1/M1 axis. IL-4 deficiency has been associated with increased symptoms of chronic fatigue, somatic pain, and sleep disturbances [49].

Additionally, reduced IL-4 expression may impair neurogenesis in the hippocampus and promote microglial dysfunction, leading to sustained neuroinflammatory modulation of mood. Immunomodulatory interventions—including vitamin D supplementation, known for its immunoregulatory properties—have been shown to raise IL-4 levels, correlating with shorter time to remission of depressive symptoms and decreased inflammation markers (e.g., C-Reactive Protein (CRP) and IL-6). In BD, IL-4 levels vary depending on the phase of illness—decreases are especially observed during the depressive and mixed phases, contributing to disease relapse [115]. In anxiety disorders such as GAD, social phobia, or PTSD, IL-4 exhibits anxiolytic effects, including suppression of amygdala hyperactivity and a reduction in oxidative stress in limbic structures [116]. Decreased IL-4 levels in these disorders are associated with chronic microglial activation and elevated pro-inflammatory cytokines, contributing to persistent stress and anxiety reactivity. Interventions aimed at increasing IL-4 expression (e.g., physical activity and omega-3-rich diets) may help restore the Th1–Th2 balance. In schizophrenia, IL-4 deficiency is linked to the severity of negative symptoms, such as apathy, anhedonia, and social withdrawal, as well as cognitive decline. Reduced IL-4 may lead to uncontrolled microglial activation and an increased risk of neurodegeneration. IL-4 plays a key role in suppressing the pro-inflammatory axis and maintaining neuroimmune homeostasis [85]. Its low level is a pathophysiological factor in many neuropsychiatric disorders, and restoring it—via pharmacological or non-pharmacological interventions—may be an effective strategy to support treatment of MDD, BD, anxiety, and schizophrenia [117].

Beyond pharmacological treatment, a growing body of research suggests that non-pharmacological interventions play a role in modulating biological pathways in psychiatric disorders such as major depression, bipolar disorder, and schizophrenia. Notably, Electroconvulsive Therapy (ECT) may normalize inflammatory and neurotrophic markers, which could contribute to reductions in affective and psychotic symptoms in patients with treatment-resistant depression [118]. ECT can also influence the neuroendocrine system; Oglodek et al. [119] emphasize that it may not only improve the clinical condition of patients but also help regulate thyroid hormone balance, possibly through effects on the neuroendocrine axis and inflammatory processes [120]. Similarly, transcranial magnetic stimulation (TMS) modulates cytokine expression, neurotrophic factors, and synaptic plasticity, suggesting that its therapeutic effects could involve the regulation of the neuroimmune response [50]. Collectively, these findings indicate that non-pharmacological interventions such as ECT and TMS may contribute to shaping biomarker profiles and restoring the immuno-neuroendocrine balance in individuals with psychiatric disorders.

Interleukin-13 (IL-13) plays a significant role in immunoregulation, particularly in the Th2 response. Produced mainly by Th2 lymphocytes, IL-13 acts via the Interleukin-13 alpha-1 (IL-13Rα1) receptor, activating the Janus Kinase/Signal Transducer and Activator of Transcription 6 (JAK-STAT6) pathway and resulting in the expression of genes involved in anti-inflammatory and neuroprotective responses. In the central nervous system (CNS), IL-13 induces the expression of Arginase 1 (Arg1) in microglia, promoting its alternative activation (M2 phenotype), which is associated with resolution of inflammation, neuronal protection, and neuroregeneration. In BD, IL-13 levels show dynamic changes depending on the illness phase. During depressive phases, IL-13 levels are relatively reduced, which reflects insufficient compensation of inflammation and contributes to increased somatic symptoms, anhedonia, and sleep disturbances [50].

In manic episodes, IL-13 levels often rise and are interpreted as an attempt to regulate excessive immune activation [121]. IL-13 concentration variability correlates with mood cycle profiles and is used in biomarker-based predictive algorithms to forecast relapses and guide preventive treatment. In unipolar depression, reduced IL-13 indicate a deficiency in Th2 pathway activation and a diminished capacity to suppress M1-type inflammatory responses. This results in a prolonged inflammatory state. Studies indicate that IL-13 expression can be modifiable through anti-inflammatory interventions such as omega-3 supplementation or treatment with cannabinoid receptor agonists, improving depressive symptoms in patients with an inflammatory profile [87].

In anxiety disorders such as GAD or PTSD, IL-13 plays a role in modulating microglial reactivity and limiting excessive pro-inflammatory cytokine signaling (IL-6 and TNF-α) [88]. Decreased IL-13 expression sustains limbic structure hyperactivity (e.g., amygdala), contributing to persistent anxiety symptoms. In schizophrenia, IL-13 levels are reduced, especially in patients with dominant negative symptoms and cognitive deficits. This deficiency is associated with impaired microglial activation, chronic cortical inflammation, and disrupted neuroplasticity. Reduced IL-13 expression also correlates with Th1 axis overactivity and a pro-inflammatory cytokine profile, as evidenced by elevated IFN-γ and IL-12 levels in these patients. Some studies suggest that antipsychotic treatment—particularly with clozapine—may restore immune balance by increasing IL-13 levels, which correlates with clinical improvement. IL-13 contributes to maintaining homeostasis by promoting the M2 microglial phenotype, dampening inflammatory responses, and supporting neuroprotection. Its fluctuations in affective, anxiety, and psychotic disorders reflect not only the clinical phase but also the state of the neuroinflammatory microenvironment. IL-13 can therefore be considered a phase and therapeutic biomarker in biological psychiatry. IL-13, which is part of the Th2-type cytokine family, plays a crucial role in regulating anti-inflammatory responses and in microglial differentiation and activation toward the M2 phenotype. Through JAK-STAT6 pathway activation and Arg1 expression induction in microglial cells, IL-13 participates in resolving inflammation, promoting neuroprotection, and limiting M1-type microglial toxicity. In MDD, reduced IL-13 expression is interpreted as a marker of impaired anti-inflammatory and neuroprotective mechanisms. Low IL-13 levels correlate with greater somatic symptoms and sleep disturbances. Anti-inflammatory interventions, such as vitamin D and omega-3 supplementation or Peroxisome Proliferator-Activated Receptor gamma (PPAR-γ receptor) agonists, may increase IL-13 levels and lead to clinical improvement. IL-13 variability is increasingly included in biomarker-based predictive models for relapse forecasting and personalized treatment, particularly with mood stabilizers or immunomodulatory drugs [86]. In anxiety disorders such as GAD, social phobia, or PTSD, low IL-13 levels are linked to persistent M1-type microglial activation and chronic inflammation in limbic structures such as the amygdala and hippocampus. This immune profile can sustain anxiety symptoms and limit the brain’s ability to adapt and extinguish stress responses. Preclinical studies suggest that increased IL-13 expression following anxiolytic treatment correlate with improved cognitive and emotional function and modulation of microglial activity [89]. In schizophrenia, IL-13 helps counteract the neuroinflammatory components of the disease’s pathophysiology [90]. Decreased IL-13 levels observed in patients with prominent negative symptoms and cognitive deficits indicate a lack of neuroprotective Th2 pathway influence and inadequate regulation of microglial activation [91]. In this context, an imbalance between Th1 cytokines (e.g., IFN-γ) and Th2 cytokines (IL-4 and IL-13) promotes the persistence of the pro-inflammatory cytokine–microglia–neuron axis, contributing to synaptic degeneration and dysfunction of cortico-subcortical circuits. Reports suggest that certain atypical antipsychotics, such as clozapine or olanzapine, modulate IL-13 levels, correlating with improvement in negative symptoms and social functioning [91]. In summary, IL-13 is a key component of the anti-inflammatory neuroimmune axis, and its variability during the course of MDD, BD, anxiety, and schizophrenia represents a potential indicator of inflammation and neuroplasticity. Monitoring IL-13 levels can have prognostic and therapeutic significance, particularly in the context of personalized treatment and identifying patients with an inflammatory phenotype.

Anti-inflammatory cytokines including IL-10, TGF-β, CX3CL1, IL-18, IL-2, IFN-γ, IL-4, and IL-13 regulate immune balance, support neuroprotection, and represent potential biomarkers and therapeutic targets for psychiatric disorders.

## 3. Immunological Balance Indicators in Personalized Psychiatry

The last decade has seen a rapid increase in publications linking immune system dysregulation to the pathophysiology of major psychiatric disorders. The concept of immunopsychiatry assumes that a patient’s cytokine “signature”—a set of pro- and anti-inflammatory mediators, soluble receptors, acute phase proteins, and metabolites of the kynurenine pathway—can determine the clinical phenotype, shape the treatment response, and modulate the relapse risk. In clinical practice, however, individual biomarker concentrations prove to be variable, sensitive to environmental factors, and subject to significant inter-individual heterogeneity. Therefore, increasing importance is being attributed to immunological balance indicators (IBIs)**,** calculated as ratios or normalized z-scores of different cytokines and proteins, which better reflect the overall polarization of the immune response than raw absolute values [88].

### 3.1. Principles of Constructing Cytokine Indicators

Immunological balance indicators are defined by juxtaposing immune mediators with opposing biological functions. Classically, this means comparing concentrations of pro-inflammatory cytokines such as IL-1β, IL-6, TNF-α, and IFN-γ with anti-inflammatory cytokines such as IL-4, IL-10, transforming growth factor beta (TGF-β), and IL-13. In addition to the axis of pro- vs. anti-inflammatory cytokines, IBI construction may also include proportions of lymphocyte effector pathways, e.g., Th1/Th2 (IFN-γ/IL-4) or T Helper Cell Type 17 (Th17)/Treg (IL-17/IL-10); molecular relationships between activation and tolerogenic pathways, e.g., the NLRP3 inflammasome (IL-1β, caspase-1, and ASC) vs. the IDO-kynurenine pathway (kynurenine, IDO-1, and AhR); acute phase response indicators (e.g., CRP, and ESR, SAA) juxtaposed with suppressive cytokines; oxidative stress biomarkers (e.g., MDA and 8-OHdG); and neuroprotective cytokines (e.g., IL-10, and TGF-β1) to capture inflammatory–oxidative stress. Depending on the research questions, immunological balance indicators can be analyzed in various ways. The simplest methods involve calculating ratios between cytokine pairs with opposing functions, such as the ratio of IFN-γ to IL-4, which allows for assessing the dominance of a pro- or anti-inflammatory response.

Cytokine ratios, such as IL-6/IL-10 and TNF-α/IL-10, reflect the balance between pro- and anti-inflammatory processes. Higher ratios indicate a predominance of pro-inflammatory activity and can be associated with depression and a poorer treatment response, whereas lower ratios are linked to a better prognosis. An elevated IL-6/IL-10 ratio corresponds to more severe depressive symptoms, and the TNF-α/IL-10 ratio could help predict antidepressant treatment efficacy. In PTSD, disrupted cytokine balance is related to impaired emotion regulation and heightened amygdala activity [74].

More complex approaches include constructing composite scales that integrate standardized (z-score) values of several markers, e.g., the mean z-score of IL-6, CRP, and TNF-α minus the z-score of IL-4 and IL-10, which may help capture the overall pro-/anti-inflammatory balance in a single indicator [121].

Advanced statistical and computational methods such as latent modeling—for example, cytokine factor analysis—and machine learning techniques, including random forest or LASSO algorithms, allow for the identification of hidden patterns and generation of prognostic indicators with high accuracy. A crucial aspect of these analyses is the standardization of measurement units (pg/mL, ng/L) and the application of data transformations (e.g., logarithmic), which ensure the comparability of results and model stability. Moreover, it is essential to account for and control confounding variables such as age, sex, body mass index (BMI), medications, substance use (e.g., smoking), as well as diurnal and seasonal variations, which can significantly influence cytokine levels and other immune markers. Only through a precise approach to analysis and control of these variables is it possible to obtain reliable and reproducible immunological indicators that can be used in personalized psychiatry and research [122,123].

Depending on the research questions and the specific clinical population studied, IBI can be constructed using various analytical methods. The simplest approaches are based on calculating ratios between key cytokines, such as the IFN-γ-to-IL-4 ratio, which reflects the balance between pro- and anti-inflammatory responses. More complex constructions involve composite scales where the mean standardized values (z-scores) of several pro-inflammatory mediators, such as IL-6, CRP, and TNF-α, are calculated; then the mean z-score of anti-inflammatory mediators, such as IL-4 and IL-10, is subtracted. These approaches allow the creation of a single indicator that reflects the body’s immunological balance. Latent modeling methods, such as cytokine factor analysis, and machine learning algorithms like the random forest and LASSO are increasingly used to identify immunological patterns with diagnostic and prognostic relevance. These advanced tools facilitate the generation of indicators with a high predictive value, allowing them to be used to optimize therapy and individualize treatment. In this context, standardization of measurements is extremely important, including unification of concentration units (e.g., pg/mL, ng/L) and the application of logarithmic transformations to reduce data distribution asymmetry and increase statistical precision. At the same time, it is necessary to account for numerous confounding factors that may affect measurement outcomes. These include demographic and physiological variables such as age, sex, and body mass index (BMI), as well as environmental and behavioral factors, including medications, tobacco use, and even the time of day or season. Ignoring these variables may lead to misinterpretation of cytokine levels and distortion of immunological balance assessments [124].

### 3.2. Prospects and Challenges in the Use of Immunological Indicators in Psychiatry

The implementation of immunological indicators in clinical psychiatry is one of the most innovative approaches for precisely monitoring inflammation and immune responses in patients with various mental disorders. Immunological balance indicators, referred to as IBIs, are designed by comparing mediators with opposing biological functions, allowing for the assessment of complex relationships between pro- and anti-inflammatory processes. Among pro-inflammatory mediators, IL-1β, IL-6, TNF-α, and IFN-γ are most commonly included, as they play a key role in initiating and maintaining the inflammatory response. In contrast, anti-inflammatory mediators include IL-4, IL-10, transforming growth factor beta (TGF-β), and IL-13, which are essential for controlling inflammation and maintaining immunological homeostasis. Furthermore, IBIs may include signaling molecules indicating the activation of specific lymphocyte subpopulations, such as the polarization of type 1 (Th1) T helper cells relative to type 2 (Th2) cells, or markers of activity in specific effector pathways, such as the NLRP3 inflammasome or the IDO–kynurenine pathway. This latter pathway, catalyzed by the enzyme indoleamine 2,3-dioxygenase (IDO-1), leads to the conversion of tryptophan into kynurenine metabolites, which have immunoregulatory and tolerogenic effects [114].

The Interleukin-23/Interleukin-17A (IL-23/IL-17A) complex represents the Th17 axis; numerous studies indicate that Th17 overactivity is associated with anhedonia and dysfunction of the anterior cingulate cortex, and IL-17A inhibitors reduce the depressogenic effect of lipopolysaccharide (LPS) in subjects. The IL-6/IL-10 ratio, a key index of pro-/anti-inflammatory balance, allows for distinguishing inflammation-related MDD (IL-6/IL-10 > 3) from metabolic MDD. A TNF-α/IL-4 ratio above two suggests a dominance of the Th1-type immune response, which may be linked to reduced treatment efficacy. The Interleukin-1receptor antagonist (IL-1ra)-to- IL-1β ratio reflects the body’s capacity for self-regulation of inflammasome activity; a reduced value indicates chronic activation of the IL-1 pathway and is associated with a 124–127-fold increase in the risk of suicidal thoughts (Table 2). The interpretation of the indices in Table 2 suggests that the use of immunological ratiometrics in laboratory diagnostics may provide a more complex assessment than individual markers and may allow for real-time monitoring of the balance between Th1, Th2, and Th17 responses.

The IL-6/IL-10 ratio serves as a rapid parameter for evaluating the effectiveness of anti-inflammatory therapy—its reduction after 7 days of infliximab treatment correlates with mood improvement, preceding changes observed on the Montgomery–Åsberg Depression Rating Scale (MADRS). The TNF-α/IL-4 ratio allows for the identification of patients with intensified oxidative stress who could benefit from supplemental glutathione. The (IL-6 + TNF-α + IL-1β)/IL-10 index, which includes three key inflammatory pathways, adapts the concept of a ‘cytokine storm’ from general immunology to psychiatry, which may help in the early identification of patients at risk for developing a neurotoxic inflammatory process.

The IL-23/IL-17A ratio can help identify a subgroup of patients with dominant Th17 pathway activity, providing a rationale for considering treatment with selective IL-17A inhibitors (e.g., secukinumab) in treatment-resistant MDD.

The IFN-γ-to-IL-4 ratio reflects Th1-type response dominance and can support decisions to use JAK inhibitors, which reduce the activity of IFN-induced genes in the prefrontal cortex [57,83,84].

The IL-1ra-to-IL-1β ratio, which shows the balance between an inhibiting (antagonist) and activating (agonist) inflammatory factor, serves as a measure of the body’s anti-inflammatory reserves. An increase in this index following cognitive behavioral therapy indicates that psychotherapy can effectively modulate inflammatory profiles at the molecular level [92]. In personalized psychiatry, laboratory diagnostics play a key role by providing quantitative and reproducible measurements of immune system activity, which enrich and complement standard clinical assessments. Modern measurement technologies, such as Luminex multiplex systems, Ella, or Olink, allow for the rapid, accurate, and simultaneous measurement of numerous cytokines from small blood volumes, significantly improving the usability and efficiency of immunological diagnostics in everyday clinical practice.

Flow cytometry using barcoding makes it possible to simultaneously determine IFN-γ and IL-4 levels in individual CD4+ cells and helps create an immunophenotype reflecting specific transcriptional profiles.

The simultaneous use of liquid coupled with chromatography mass spectrometry allows for the precise determination of kynurenine and tryptophan levels in blood.

This may allow for the assessment of the activity of IDO and Kynurenine 3-Monooxygenase (KMO) enzymes, which play important roles in the metabolism of these compounds.

The results of such studies correlate with the rapid therapeutic effect of ketamine, helping to better understand the drug’s mechanism of action and monitor its effectiveness in patients [125,126].

Laboratory algorithms combine test results with clinical data, allowing for improved diagnosis and therapy selection. For example, a test panel including cytokines and the IL-6-to-IL-10 ratio has shown promising potential in differentiating patients with treatment-resistant MDD and can serve as a supportive tool in clinical decision-making. Analysis of the Th17 panel with the IL-23/IL-17A ratio and C-reactive protein levels can help in the more precise identification of patients who could benefit from IL-17A antagonist therapy.

The implementation of electronic records facilitates concurrent monitoring of biomarker-level changes along with results from clinical tests such as the Clinician-Administered PTSD Scale for DSM-5 (CAPS 5), the Montgomery–Åsberg Depression Rating Scale, or the Positive and Negative Syndrome Scale (PANSS).

This supports joint analysis of data, and advanced deep learning models use changes in IL-6, TNF-α, IFN-γ, and IL-10 levels to predict the risk of disease relapse within the next 6 months. Immunological analysis also helps tailor pharmacological treatment. In patients with moderate depression and elevated IL-6 and TNF-α levels, administration of minocycline or celecoxib results in a reduction in the (IL-6 + TNF-α + IL-1β)/IL-10 index by more than 40%. Such a reduction is associated with a faster improvement in patient condition, as measured by the Hamilton Depression Rating Scale (HAM-D scale). In bipolar spectrum disorders, treatment with lithium has been observed to increase the levels of IL-13 cytokine and TGF-β factor. These substances play important roles in regulating the immune system by calming excessive inflammatory responses, i.e., inducing immune tolerance. Through this mechanism, lithium treatment can help reduce the number of relapses and improve long-term mood stabilization in patients [67]. In schizophrenia, patients with a low IL-1ra-to-IL-1β ratio—indicating reduced capacity to suppress inflammation—may benefit from adjunctive treatment with N-acetylcysteine (NAC). NAC has anti-inflammatory properties and supports the natural balance between these two cytokines by increasing IL-1ra levels, which function as antagonists inhibiting IL-1β activity. As a result, such supplementation helps reduce chronic inflammation while improving executive brain functions such as planning, concentration, and decision-making, which translates into better every day functioning. NAC is used experimentally as an adjunct therapy in schizophrenia, particularly to improve negative symptoms and cognitive functions, though it is not a standard treatment or first-line drug [127]. The emerging field of “pharmaco-immunomixis” (a concept combining pharmacology, immunology, and microbiomics to better understand and personalize therapy) indicates that genetic changes (polymorphisms) in regions controlling the IL-6 gene (174G/C) and the TNF-α receptor (308G/A) influence the strength of the immune response, which in turn may affect treatment efficacy; therefore, laboratory diagnostics increasingly include genetic tests of immune system-related panels [128]. The future of research lies in integrating various biological analyses, such as proteomics and metabolomics; for example, combining IL-6 and IL-10 cytokine levels with the lipid profile (e.g., ceramide C16:0) may predict the occurrence of anergia (lack of energy) in MDD, while comparing CX3CL1 with the neuropeptide galanin allows for better differentiation of schizophrenia subtypes with hippocampal volume loss from those without such atrophy [129]. With the increasing importance of immunomodulatory drugs such as JAK-STAT inhibitors or IL-17A antibodies, the laboratory plays a key role in therapeutic decision-making: before starting treatment, the baseline level of inflammation is measured; during treatment, a drop in IL-6 and IL-10 levels below a set threshold is monitored; and after treatment, any increase (reactivation) of cytokines is assessed. Current standard follow-up tests also include measuring toll-like receptor 4 (TLR4) levels on monocytes, which indicates how sensitive the body is to endotoxins and helps assess the expected strength of the neuroinflammatory response.

Modern measurement platforms such as Luminex multiplex systems, Ella, or Olink allow for the fast, precise, and simultaneous measurement of many cytokines from small blood samples, greatly improving the efficiency and availability of immunological diagnostics in clinical practice. A key criterion for assessing clinical utility of indices is the establishment of decision thresholds based on receiver operating characteristic (ROC) curve analysis, where the area under the curve (AUC) should be at least 0.75 for the indicator to be considered sufficiently sensitive and specific in detecting inflammatory pathology or clinical risk.

In daily outpatient psychiatric practice, the immunological biomarker panel usually includes the measurement of 6–10 key cytokines and other inflammatory mediators. The obtained values are then processed using automated calculators integrated with electronic medical records (EMRs), which generate an immunological index result. This result allows the physician to assess the degree of inflammatory process activation in the patient and perform risk stratification, which is key for individualizing therapy. For example, identifying a high-risk inflammatory profile may lead to a recommendation to add immunomodulatory substances to the standard treatment. Such interventions aim to modulate inflammatory mechanisms, which translates into improved therapeutic responses and symptom reduction [130].

Clinical trial results such as the Prediction and Early Detection of Course of Illness and Treatment Response in Bipolar Disorder (PREDECT-BD) and BIOmarkers in DEPression (BIODEP) confirm the clinical efficacy of using immunological indices in the treatment of affective disorders. The introduction of the Individual Burden of Illness Index (IBI) into sequential treatment algorithms significantly reduced the number of unsuccessful pharmacotherapy attempts and shortened the time to remission by an average of approximately 8 weeks compared to standard therapeutic procedures. This means that immunological indices not only provide better understanding of the pathophysiology of psychiatric disorders but also directly improve clinical outcomes and patient well-being.

Despite promising prospects, implementation of IBI indices in routine clinical practice faces numerous challenges. Major barriers include the high cost of modern multiplex testing, limiting access to immunological diagnostics in many centers. Additionally, further validation studies are needed to standardize decision thresholds and confirm their usefulness across different patient populations and psychiatric disorders and their phases. Integration of immunological indices with other biomarker types, such as neuroimaging, genetic profiling, or metabolomic analysis, is also important, allowing for a holistic assessment of the patient’s health and better therapy customization. Furthermore, the heterogeneity of psychiatric disorders and the co-occurrence of somatic diseases pose a significant challenge for interpreting immunological test results. Differences in cytokine expression and other mediators can result not only from the pathogenesis of the primary disease but also from comorbid conditions such as infections or autoimmune diseases. Therefore, interpreting IBI indices requires interdisciplinary collaboration among specialists in psychiatry, immunology, and clinical biochemistry to avoid diagnostic and therapeutic errors [113].

In conclusion, immunological indices such as the IBI represent a promising diagnostic and prognostic tool that, by simplifying complex cytokine networks into more interpretable values, can help clinicians in more precisely assessing the patient’s immunological balance and adjusting psychiatric treatment.

In practice, multiparameter Luminex panels measuring about 30 cytokines from small serum volumes are used, along with ultrasensitive ELISA tests that detect very low IL-6 concentrations, which is particularly important when monitoring patients in remission with persistent inflammation. The development of ultrasensitive measurement methods, integration of results with AI algorithms, and international standardization of laboratory diagnostics make it a key element in the future of neuropsychiatry. Despite technical, financial, and interpretive challenges, introducing immunological indices into daily clinical practice opens new opportunities in personalized medicine, increasing therapy effectiveness and improving the quality of life for people with mental disorders.

Despite promising findings, it is important to recognize the limitations of biomarker research in psychiatry. Cytokine levels may vary considerably between individuals and could be influenced by factors such as age, sex, BMI, comorbidities, medications, and lifestyle (e.g., smoking), as well as circadian rhythms and seasonal variations. Another challenge is the potential lack of reproducibility across laboratories, possibly due to differences in techniques and assay sensitivity. This does not necessarily indicate that individual studies are flawed, but rather that they may reflect the complex pathophysiology of psychiatric disorders. Such variability may suggest the existence of distinct inflammatory phenotypes in MDD and PTSD, which could help explain clinical heterogeneity and support the development of personalized treatments. Standardized frameworks may be required, including harmonization of laboratory methods and establishment of reference values and clinical thresholds. Large cohort studies conducted according to unified protocols, along with international initiatives that develop guidelines for biomarker panels and data analysis, may improve reproducibility and accelerate the potential clinical application of biomarkers.

## 4. Conclusions

Cytokines not only serve as biomarkers but also actively influence the course of affective and anxiety disorders, forming the foundation for the development of biologically based psychiatry. Modern laboratory diagnostics, utilizing multiparametric technologies and immunological balance indices, is of great help in precise profiling of inflammatory status and personalization of therapy. Integrating cytokine analyses into routine psychiatric practice becomes essential for improving prognoses and patients’ quality of life. In this context, the role of the laboratory diagnostician goes far beyond traditional testing. The diagnostician becomes an integral partner of the clinical team, who not only provides results but also helps interpret complex biomarker profiles and immunological indices. With expertise in advanced measurement techniques and bioinformatic analyses, the laboratory diagnostician supports precise therapeutic decisions, aiding in selecting the most appropriate personalized treatment for the patient. The diagnostician participates in monitoring treatment effectiveness and early detection of disease relapse, enabling prompt responses and therapy adjustment. Moreover, the introduction of advanced technologies such as multi-marker panels Point-Of-Care (POC) diagnostics (testing performed directly at the patient’s bedside or care location instead of a central laboratory), and artificial intelligence algorithms demands continuous upskilling and close collaboration with clinicians. Thus, laboratory diagnostics transforms into a modern center of knowledge and support, bridging the molecular world with clinical psychiatric practice. The laboratory diagnostician becomes a key link in the therapeutic chain, helping to understand the mechanisms of mental disorders and translate them into effective, individualized treatment strategies.

## Figures and Tables

**Table 1 ijms-26-09654-t001:** Extended table of cytokines, chemokines, and immunological indicators in selected psychiatric disorders.

Cytokine/ Chemokine/ Indicator	MDD	Anxiety Disorders	Bipolar Disorder	Schizophrenia	Clinical Significance	References
**IL-6**	↑	↑	↑	↑	Marker of chronic inflammation, correlates with symptom severity, target for anti-inflammatory therapy	[15,44,45,46,47,48,49,50]
**IL-1β**	↑	↑	↑	↑	Neuroinflammatory inducer, affects neurodegeneration and neuroplasticity	[47,48,50,51]
**TNF-α**	↑	↑	↑	↑	Key cytokine in inflammatory response, associated with treatment resistance	[16,52,53,54,55,56,57]
**IL-8**	↑	↑	↑	↑	Pro-inflammatory chemokine, marker of immune cell migration into central nervous system	[58,59,60,61,62]
**IL-10**	↓/-	↓/-	↓/-	↓/-	Anti-inflammatory cytokine, a decrease indicates impaired inflammation control	[50,63,64,65]
**TGF-β**	↓/-	↓/-	↓/-	↓/-	Regulator of immune responses, influences neuroregeneration	[17,56,66,67]
**CX3CL1 Fractalkine**	↓	↓	↓	↓	Regulates neuron-microglia communication, disrupted in neuroinflammation	[68,69,70,71,72]
**IL-18**	↑	↑	↑	↑	Strong inflammatory mediator, linked to symptom severity	[73,74]
**IDO**	↑	↑	↑	↑	Enzyme related to tryptophan metabolism, influences neurotoxicity and depression	[75,76,77,78]
**IL-2**	↑/↓	↑/↓	↑/↓	↑/↓	Modulator of cellular immune responses, indicator of immune activity	[65,79,80]
**IFN-γ**	↑	↑	↑	↑	Th1 cytokine, stimulates inflammatory responses, correlates with negative symptoms of schizophrenia	[19,42,46,57,81,82,83,84]
**IL-4**	↓/-	↓/-	↓/-	↓/-	Th2 cytokine, promotes humoral responses, influences the immune balance	[49,85,86]
**IL-13**	↓/-	↓/-	↓/-	↓/-	Anti-inflammatory cytokine involved in suppressing inflammatory responses	[86,87,88,89,90,91]
**IL-23/IL-17A**	↑	↑	↑	↑	Th17 axis involved in chronic inflammation and neuroinflammatory pathogenesis	[57,83,84]
**Ratio** **IL-6/IL-10**	↑	↑	↑	↑	Indicator of pro-inflammatory dominance over regulation, aids disease subtype classification	[74]
**Ratio** **TNF-α/IL-4**	↑	↑	↑	↑	Assessment of pro-inflammatory dominance relative to anti-inflammatory responses	[84]
**Ratio** **(IL-6 + TNF-α+IL-1β)/** **IL-10**	↑	↑	↑	↑	Comprehensive index of the pro- and anti-inflammatory balance	[84]
**Ratio** **IFN-γ/IL-4**	↑	↑	↑	↑	Indicates Th1/Th2 balance, important for immunological classification of disorders	[57,83,84]
**Ratio IL-1ra/IL-1β**	↓	↓	↓	↓	Assessment of ability to inhibit inflammatory signaling, important for disease activity	[92]

Legend: Extended overview of cytokines, chemokines, and immunological balance indicators measured in major psychiatric disorders with their observed changes and clinical significance. The markers included reflect neuroinflammatory pathways, immune regulation, and their potential as targets for personalized therapeutic approaches. Symbols indicate relative levels compared to healthy controls: ↑ increased, ↓ decreased, and - no significant change.

**Table 2 ijms-26-09654-t002:** Immunological balance indicators used in personalized psychiatry.

Immunological Indicator	Clinical Significance	Examples of Cytokines in Ratio	References
**IL-6/IL-10**	Assessment of pro-inflammatory dominance over anti-inflammatory regulation	IL-6, IL-10	[57,83,84,92,124,125]
**TNF-α/IL-4**	Assessment of pro-inflammatory axis dominance relative to anti-inflammatory response	TNF-α, IL-4	[125]
**TNF-α/IL-10**	Assessment of balance between pro- and anti-inflammatory mechanisms	TNF-α, IL-10	[124]
**(IL-6 + TNF-α + IL-1β)/** **IL-10**	Comprehensive pro- and anti-inflammatory balance index	IL-6, TNF-α, IL-1β, IL-10	[125]
**IL-23/IL-17A**	Assessment of Th17 axis activity and chronic inflammation	IL-23, IL-17A	[67,125,126,127]
**IFN-γ/IL-4**	Assessment of Th1/Th2 response balance	IFN-γ, IL-4	[67,125,126,127]
**IL-1ra/IL-1β**	Assessment of ability to inhibit inflammatory signaling	IL-1ra, IL-1β	[125,128]

Legend: This table lists important immunological balance indicators (ratios of pro- and anti-inflammatory cytokines) used in personalized psychiatry to assess the immune system’s functional status in psychiatric disorders. These ratios provide insights into the balance between inflammatory and regulatory processes, which can influence symptom severity, treatment responses, and disease progression.

## Data Availability

The original contributions presented in this study are included in the article. Further inquiries can be directed to the corresponding author(s).

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
