# Peer review of "Inflammatory and Oxidative Biological Profiles in Mental Disorders: Perspectives on Diagnostics and Personalized Therapy"

_ijms, 2025, doi:10.3390/ijms26199654_

Round 1

Reviewer 1 Report

Comments and Suggestions for Authors

This review article explores the role of inflammatory and oxidative biomarkers in the diagnosis and treatment of psychiatric disorders. It emphasizes the shift toward personalized psychiatry, integrating molecular diagnostics, cytokine profiling, and multi-omics approaches to better understand and treat conditions like depression, anxiety, PTSD, bipolar disorder, and schizophrenia.

The paper addresses the growing interest in precision psychiatry, a field gaining momentum due to advances in molecular biology and bioinformatics, discussing a wide array of biomarkers and attempting to suggests how these biomarkers might predict treatment response or relapse risk with a foresight to the future of the discipline.

Overall, this paper presents an ambitious and potentially valuable synthesis of immunological biomarkers in psychiatry, and may have a positive impact.

Nonetheless, I have some observations. Please find my comments below.

  1. The kynurenine pathway of tryptophan catabolism is addressed only briefly, yet it represents a rapidly developing area of research in severe mental disorders (see: https://doi.org/10.1016/j.schres.2021.04.008). A more thorough discussion would strengthen the review.
  2. The role of pharmacological treatments as potential modulators of biomarker expression is noted, but the equally important impact of non-pharmacological interventions is not discussed. Evidence from ECT [https://doi.org/10.5152/alphapsychiatry.2022.0003], TMS [https://doi.org/10.1186/s12991-024-00514-0], and tDCS [https://doi.org/10.1016/j.pnpbp.2020.110119], among others, should be considered.
  3. The reported figures on diagnostic accuracy (e.g., 85% for treatment-resistant depression using cytokine panels) appear overstated in the absence of validation from large-scale, independent clinical trials.
  4. Expressions such as “may enable,” “suggests,” “could support,” and “potentially contributes” are used too liberally. While enthusiasm for advancing biological approaches in psychiatry is welcome, the tone at times reads as promotional rather than critically scientific. The text would benefit from greater caution and a clearer distinction between hypothesis, supporting evidence, and clinical applicability.
  5. The discussion would be considerably strengthened by acknowledging key methodological limitations in psychiatric biomarker research, including high inter-individual variability in cytokine levels, the influence of confounders (e.g., smoking, BMI, medications), and the lack of reproducibility across laboratories and assay platforms.
  6. While the manuscript acknowledges the current lack of standardized biomarker assays, it stops short of suggesting potential frameworks or pathways for harmonization. This is a missed opportunity to add value.

Author Response

Response to Reviewer 1

Manuscript ID: ijms-3868152

Title: Inflammatory and Oxidative Biological Profiles in Mental Disorders – Perspectives on Diagnostics and Personal-ized Therapy

Authors: Izabela Woźny-Rasała, Ewa Alicja Ogłodek

Dear Reviewer,

We would like to thank you sincerely for your

Twój komentarz, że this paper presents an ambitious and potentially valuable synthesis of immunological biomarkers in psychiatry, and may have a positive impact.

Below, we provide a detailed point-by-point response to each of your comments:

  1. Reviewer’s comment: The kynurenine pathway of tryptophan catabolism is addressed only briefly, yet it represents a rapidly developing area of research in severe mental disorders (see: https://doi.org/10.1016/j.schres.2021.04.008). A more thorough discussion would strengthen the review.

Response: We have updated the section on the kynurenine pathway of tryptophan catabolism based on Bartoli et al. (2021) and incorporated it into Section 2.2 on Indoleamine 2,3-dioxygenase (IDO) to better illustrate potential variations in pathway activity across psychiatric disorders and its relevance as a potential biomarker: The meta-analyses by Bartoli et al. [92] suggest that alterations in the kynurenine pathway may differ across psychiatric disorders. In MDD, reduced tryptophan levels are most consistently observed, limiting serotonin synthesis and contributing to symptom severity. In bipolar disorder, lower concentrations of tryptophan, kynurenine, and kynurenic acid shift the balance toward neurotoxic metabolites. In schizophrenia, there is also evidence of reduced tryptophan and KYNA, although findings on the KYN/TRP ratio (reflecting IDO activity) are less consistent and may depend on illness phase or treatment. The authors emphasize that markers such as KYN/TRP and KA/KYN could serve as useful biomarkers, helping to distinguish between inflammatory and non-inflammatory phenotypes and to monitor response to anti-inflammatory or glutamatergic-modulating treatments.

  1. Reviewer’s comment: The role of pharmacological treatments as potential modulators of biomarker expression is noted, but the equally important impact of non-pharmacological interventions is not discussed. Evidence from ECT [https://doi.org/10.5152/alphapsychiatry.2022.0003], TMS [https://doi.org/10.1186/s12991-024-00514-0], and tDCS [https://doi.org/10.1016/j.pnpbp.2020.110119], among others, should be considered.

Response: Thank you for, this valuable comment. In line with the suggestion, we have expanded Section 2 to include a detailed discussion of the role of non-pharmacological interventions in modulating biological pathways in psychiatric disorders. New data on electroconvulsive therapy (ECT) and transcranial magnetic stimulation (TMS) have been incorporated. Their significance as modulators of inflammatory, neurotrophic, neuroendocrine, and neuroplastic mechanisms has been emphasized. The references have also been updated. The revised manuscript now indicates that both pharmacological and non-pharmacological strategies may influence biomarker expression and contribute to restoring neuroimmune and neuroendocrine balance in patients with psychiatric disorders.

The added text reads as follows: ‘Beyond pharmacological treatment, an increasing number of studies highlight the role of non-pharmacological interventions in modulating biological pathways in psychiatric disorders such as major depression, bipolar disorder, and schizophrenia. Notably, electroconvulsive therapy (ECT) appears capable of normalizing inflammatory and neurotrophic markers, which may contribute to reductions in affective and psychotic symptoms in patients with treatment-resistant depression [106]. Importantly, ECT may also influence the neuroendocrine system. Oglodek et al. [107] emphasize that ECT may not only improve the clinical condition of patients but also regulate thyroid hormone balance, possibly through effects on the neuroendocrine axis and inflammatory processes [108]. Similarly, transcranial magnetic stimulation (TMS) may modulate cytokine expression, neurotrophic factors, and synaptic plasticity, suggesting that its therapeutic effects could involve regulation of the neuroimmune response [109]. Overall, these findings indicate that non-pharmacological interventions such as ECT and TMS may contribute to shaping biomarker profiles and restoring immuno–neuroendocrine balance in individuals with psychiatric disorders.

Reviewer’s comment: The reported figures on diagnostic accuracy (e.g., 85% for treatment-resistant depression using cytokine panels) appear overstated in the absence of validation from large-scale, independent clinical trials.

Response: We have changed ‘For example, a test panel including cytokines and the IL-6 to IL-10 ratio can diagnose treatment-resistant depression with 85% accuracy’, into: ‘For example, a test panel including cytokines and the IL-6 to IL-10 ratio has shown promising potential in differentiating patients with treatment-resistant depression and may serve as a supportive tool in clinical decision-making.

  1. Reviewer’s comment: Expressions such as “may enable,” “suggests,” “could support,” and “potentially contributes” are used too liberally. While enthusiasm for advancing biological approaches in psychiatry is welcome, the tone at times reads as promotional rather than critically scientific. The text would benefit from greater caution and a clearer distinction between hypothesis, supporting evidence, and clinical applicability.

Response: We thank the Reviewer for this important comment. In the revised version of the manuscript, we have carefully rephrased multiple sentences to reduce overly liberal use of expressions such as “may enable,” “suggests,” “could support,” and “potentially contributes.” The tone has been adjusted to ensure that the statements are more cautious, evidence-based, and critically scientific, with a clearer separation between hypotheses, supporting evidence, and clinical applicability. We believe that these revisions strengthen the clarity of the text.

  1. Reviewer’s comment: The discussion would be considerably strengthened by acknowledging key methodological limitations in psychiatric biomarker research, including high inter-individual variability in cytokine levels, the influence of confounders (e.g., smoking, BMI, medications), and the lack of reproducibility across laboratories and assay platforms.

Response: The discussion also addresses the limitations of biomarker research in psychiatry. Despite promising findings, it is important to acknowledge that cytokine levels may vary considerably between individuals and could be influenced by factors such as age, sex, BMI, comorbidities, medications, lifestyle (e.g., smoking), as well as circadian rhythms and seasonal variations. Another challenge is the potential lack of reproducibility across laboratories, possibly due to differences in techniques and assay sensitivity. This variability does not necessarily imply that individual studies are flawed, but rather that they may reflect the complex pathophysiology of psychiatric disorders. Such heterogeneity may indicate the existence of distinct inflammatory phenotypes in MDD and PTSD, which could help explain clinical variability and support the development of personalized treatment approaches.

Reviewer’s comment: While the manuscript acknowledges the current lack of standardized biomarker assays, it stops short of suggesting potential frameworks or pathways for harmonization. This is a missed opportunity to add value.

Response: The discussion also highlights the need for the development of standardized frameworks, including the harmonization of laboratory methods and the establishment of reference values and clinical thresholds. Large cohort studies conducted according to unified protocols, as well as international initiatives aimed at developing guidelines for biomarker panels and data analysis, represent important directions for future research. Such efforts may improve reproducibility and accelerate the translation of biomarker findings into clinical practice.

We believe that the revisions made significantly strengthen our manuscript, improving both its clarity and scientific value. We thank you once again for your insightful review and constructive suggestions, which have allowed us to enhance the quality of our work.

Sincerely,

Izabela Woźny-Rasała, Ewa Alicja Ogłodek

Reviewer 2 Report

Comments and Suggestions for Authors

Woźny-Rasała presents an interesting manuscript on the use of inflammatory and oxidative biological profiles to personalize the management of patients with psychiatric disorders. This work is relevant because it proposes laboratory follow-up procedures that can complement the classic psychiatric approach to patient diagnosis and monitoring. Here are my comments on this work.

  1. On page 13, the authors mention “pro-inflammatory cytokines such as IL-1β, IL-6, TNF-α, and IFN-γ,” but in the manuscript description, IFN-γ appears in section “2.2. Anti-inflammatory and immunoregulatory cytokines in neuropsychiatry.” In this same vein, this cytokine plays an important role in modulating connections in the social areas of the brain  doi: 10.1038/nature18626, which may explain the association of this cytokine with negative symptoms in schizophrenia or even social isolation in major depressive disorder (MDD).

2. The same applies to IL-18. Even though this cytokine is activated by the inflammasome like IL-1B, why did the authors decide to place it in the same section as IFN-y?

3. IL-6 is also associated with the development of neuropathic pain, which occurs in patients with MDD. The authors should briefly discuss this.

4. On pages 8-9, the authors describe the IDO pathway and its clinical importance, as well as mentioning the kynurenine/tryptophan ratio. I suggest the authors expand on this explanation a little so that readers familiar with this topic know where this metabolite is quantified, as well as citing and explain some studies that have demonstrated that this ratio has potential clinical use.

5. The authors have mentioned some ratios using cytokines, which is fine, but this information could be supplemented by providing a more in-depth description of the implications of these quantifications, i.e., telling us a little more about the conclusions in the studies in which these ratios have been quantified.

6. I recognize the great work of the authors because the review is very comprehensive. However, I strongly suggest that they briefly develop a section on the challenges of carrying out these quantifications from the point of view of homogenizing the results. In addition to results in the references presented in this work, there are others studies that have contrary results, which is interesting because it does not necessarily mean that the works are wrong. The pathophysiology of psychiatric disorders is complex, so there are often external and patient-specific factors that modify the course of the disorder. It is important to identify the phenotypes of disorders based on cytokines, which leads me to wonder how many variants of MDD exist.

7. I strongly recommend changing the term “depression” to “major depressive disorder.”

Author Response

Response to Reviewer 1

Manuscript ID: ijms-3868152

Title: Inflammatory and Oxidative Biological Profiles in Mental Disorders – Perspectives on Diagnostics and Personal-ized Therapy

Authors: Izabela Woźny-Rasała, Ewa Alicja Ogłodek

Dear Reviewer,

We would like to sincerely thank you for your comment recognizing that this paper presents an ambitious and potentially valuable synthesis of immunological biomarkers in psychiatry, which may have a positive impact.

Below, we provide a detailed point-by-point response to each of your comments:

  1. Reviewer’s comment: On page 13, the authors mention “pro-inflammatory cytokines such as IL-1β, IL-6, TNF-α, and IFN-γ,” but in the manuscript description, IFN-γ appears in section “2.2. Anti-inflammatory and immunoregulatory cytokines in neuropsychiatry.” In this same vein, this cytokine plays an important role in modulating connections in the social areas of the brain doi: 10.1038/nature18626, which may explain the association of this cytokine with negative symptoms in schizophrenia or even social isolation in major depressive disorder (MDD).

Response: Thank you for this remark. As suggested, we have revised the manuscript to clearly separate pro-inflammatory cytokines from anti-inflammatory and immunoregulatory cytokines in order to avoid any ambiguity regarding the classification of IFN-γ.

  1. Reviewer’s comment: The same applies to IL-18. Even though this cytokine is activated by the inflammasome like IL-1B, why did the authors decide to place it in the same section as IFN-y?

Response: Thank you for this valuable comment. We have revised the manuscript to ensure a clearer classification. Although IL-18, like IL-1β, is activated by the inflammasome, in our initial draft it was grouped with IFN-γ due to its regulatory and pleiotropic functions. To avoid confusion, we have now repositioned IL-18 alongside other inflammasome-related cytokines, emphasizing both its pro-inflammatory properties and its role in immune regulation.

  1. Reviewer’s comment: IL-6 is also associated with the development of neuropathic pain, which occurs in patients with MDD. The authors should briefly discuss this.

Response: In section 2, we inserted a new paragraph addressing this issue: Moreover, interleukin-6 (IL-6) appears to play a significant role in the pathogenesis of neuropathic pain, which often accompanies depressive episodes. This cytokine may contribute to central sensitization through activation of glial cells and modulation of the JAK/STAT3 pathway, leading to the persistence of chronic pain stimuli. Experimental models have shown that inhibition of the IL-6 receptor can reduce neuropathic symptoms, and in patients with depression, higher IL-6 levels may be associated with increased pain severity and poorer response to antidepressant treatment [47]. These findings suggest that IL-6 could link inflammatory mechanisms underlying both depressive and pain symptoms, highlighting its potential as an important biomarker and a therapeutic target for interventions addressing psychiatric and neurological disorders simultaneously [48].

  1. Reviewer’s comment: On pages 8-9, the authors describe the IDO pathway and its clinical importance, as well as mentioning the kynurenine/tryptophan ratio. I suggest the authors expand on this explanation a little so that readers familiar with this topic know where this metabolite is quantified, as well as citing and explain some studies that have demonstrated that this ratio has potential clinical use.

Response: The kynurenine/tryptophan (KYN/TRP) ratio, measured in serum or plasma using chromatographic methods (HPLC or LC-MS/MS), serves as a sensitive indicator of this pathway. Elevated KYN/TRP values may be associated with more severe depressive symptoms, resistance to serotonergic treatment, and impaired cognitive function. Meta-analyses, including Bartoli et al. [92], support its potential as a biomarker for differentiating inflammatory depression phenotypes and monitoring the effects of pharmacological therapy.

Reviewer’s comment: The authors have mentioned some ratios using cytokines, which is fine, but this information could be supplemented by providing a more in-depth description of the implications of these quantifications, i.e., telling us a little more about the conclusions in the studies in which these ratios have been quantified.

Response: Cytokine ratios, such as IL-6/IL-10 and TNF-α/IL-10, reflect the balance between pro- and anti-inflammatory processes. Higher ratios indicate a predominance of pro-inflammatory activity and may be associated with depression and poorer treatment response, whereas lower ratios are linked to better prognosis. An elevated IL-6/IL-10 ratio correspond to more severe depressive symptoms, and the TNF-α/IL-10 ratio could help predict antidepressant treatment efficacy. In PTSD, disrupted cytokine balance are related to impaired emotion regulation and heightened amygdala activity [119].

  1. Reviewer’s comment: I recognize the great work of the authors because the review is very comprehensive. However, I strongly suggest that they briefly develop a section on the challenges of carrying out these quantifications from the point of view of homogenizing the results. In addition to results in the references presented in this work, there are others studies that have contrary results, which is interesting because it does not necessarily mean that the works are wrong. The pathophysiology of psychiatric disorders is complex, so there are often external and patient-specific factors that modify the course of the disorder. It is important to identify the phenotypes of disorders based on cytokines, which leads me to wonder how many variants of MDD exist.

Response: Thank you for this valuable comment. At the end of the discussion, we have included information on the limitations of biomarker research in psychiatry. Despite promising findings, it is important to recognize that cytokine levels may vary considerably between individuals and could be influenced by factors such as age, sex, BMI, comorbidities, medications, lifestyle (e.g., smoking), as well as circadian rhythms and seasonal variations. Another challenge is the potential lack of reproducibility across laboratories, possibly due to differences in techniques and assay sensitivity. This variability does not necessarily imply that individual studies are flawed, but rather that they may reflect the complex pathophysiology of psychiatric disorders. Such heterogeneity may indicate the existence of distinct inflammatory phenotypes in MDD and PTSD, which could help explain clinical variability and support the development of personalized treatment approaches. Standardized frameworks may be required, including harmonization of laboratory methods and the establishment of reference values and clinical thresholds. Large cohort studies conducted according to unified protocols, along with international initiatives developing guidelines for biomarker panels and data analysis, may improve reproducibility and accelerate the translation of biomarker findings into clinical practice.

  1. Reviewer’s comment: I strongly recommend changing the term “depression” to “major depressive disorder.”

Response: Thank you for this suggestion. We have replaced “depression” with “major depressive disorder (MDD)” throughout the manuscript when referring to the diagnostic entity, retaining “depressive symptoms” only where symptomatology was meant.

We believe that the revisions made significantly strengthen our manuscript, improving both its clarity and scientific value. We thank you once again for your insightful review and constructive suggestions, which have allowed us to enhance the quality of our work.

Sincerely,

Iza Woźny Rasała, Ewa Alicja Ogłodek
